ecology/microbiology/health and disease and epidemiology

*Leishmania*, Phlebotominae, cutaneous leishmaniasis, feeding sources, ecology

**Author for correspondence:**
Juan David Ramírez
e-mail: juand.ramirez@urosario.edu.co

†Both authors contributed as first authors.

# Complex ecological interactions across a focus of cutaneous leishmaniasis in Eastern Colombia: novel description of *Leishmania* species, hosts and phlebotomine fauna

Claudia M. Sandoval-Ramírez[1,†], Carolina Hernández[2,†],
Aníbal A. Teherán[3], Reinaldo Gutierrez-Marin[4],
Ruth A. Martínez-Vega[5], Duvan Morales[2],
Richard Hoyos-Lopez[6], Astrid Araque-Mogollón[7]
and Juan David Ramírez[2]

[1]Grupo de Investigaciones en Ciencias Básicas y Aplicadas para la Sostenibilidad (CIBAS), Facultad de Ciencias Exactas, Naturales y Agropecuarias, Universidad de Santander, Bucaramanga, Colombia
[2]Grupo de Investigaciones Microbiológicas-UR (GIMUR), Departamento de Biología, Facultad de Ciencias Naturales, Universidad del Rosario, Bogotá, Colombia
[3]Grupo de Investigación COMPLEXUS, Fundación Universitaria Juan N, Corpas, Bogotá, Colombia
[4]Grupo de Investigación en Enfermedades Tropicales e Infecciosas (GIEPATI), Universidad de Pamplona, Pamplona, Colombia
[5]Grupo de Investigación Salud Comunidad-UDES, Universidad de Santander, Bucaramanga, Colombia
[6]Grupo de Investigación en Enfermedades Tropicales y Resistencia Bacteriana, Universidad del Sinú, Montería, Colombia
[7]Instituto Departamental de Salud, Laboratorio Departamental de Salud, Norte de Santander, Cúcuta, Colombia

JDR, 0000-0002-1344-9312

This study aimed to analyse the patterns of diversity, blood sources and *Leishmania* species of phlebotomines in a focus of cutaneous leishmaniasis in Arboledas, Eastern Colombia. In total, 1729 phlebotomines were captured in two localities (62.3% Siravita and 37.7% Cinera) and five environments of Norte de Santander. We identified 18 species of phlebotomines:

*Pintomyia ovallesi* (29.8%), *Psychodopygus davisi* (20.3%), *Pi. spinicrassa* (18.5%) and *Lutzomyia gomezi* (15.8%) showed the highest abundance. Species diversities were compared between Cinera (15.00) and Siravita (20.00) and among five microenvironments: forest remnants (19.49), coffee plantations (12.5), grassland (12.99), cane plantations (11.66) and citrus plantations (12.22). *Leishmania* DNA was detected in 5.8% (80/1380) of females, corresponding mainly to *Pi. ovallesi* (22/80; 27.2%), *Lu. gomezi* (17/80; 21.3%) and *Pi. spinicrassa* (11/80; 13.8%). *Leishmania* species were 63.1% *L. braziliensis*, 18.5% *L. panamensis*, 13.2% *L. infantum* and 6.1% *L. amazonensis*. The most frequent feeding sources were *Homo sapiens* (50%), *Bos taurus* (13.8%) and *Canis lupus familiaris* (10.3%). This focus of cutaneous leishmaniasis has a high diversity of *Leishmania*-carrying phlebotomines that feed on domestic animals. The transmission of leishmaniasis to human hosts was mainly associated with *Lu. gomezi*, *Pi. ovallesi* and *L. braziliensis*.

# 1. Introduction

Leishmaniasis is a disease caused by protozoan parasites that are transmitted by insect vectors of the Psychodidae family. The disease represents a serious global public health problem despite multiple mitigation efforts, including surveillance systems and public health interventions. Leishmaniasis is present in at least 98 countries, three territories and five continents worldwide. There are currently around 12 million people in the world with this infection, and approximately 2 million new cases of leishmaniasis occur each year [1]. In the Americas, cutaneous and mucosal leishmaniasis constitute serious public health problems; they are endemic to 22 countries in which species of the *Viannia* subgenus are responsible for the majority of cutaneous leishmaniasis (CL) cases [2].

In Colombia, leishmaniasis is endemic throughout almost the entire national territory, and over 98% of cases are CL. It is estimated that more than 12 million people are at risk of acquiring the infection, with Colombia having the second-highest incidence in the Americas after Brazil [1,3]. Colombia and the inter-Andean valleys have leishmaniasis foci, and working in coffee plantations in these regions appears to be a risk factor for acquiring the disease [4]. Outbreaks of CL have been reported since 1984 and are caused by *Leishmania braziliensis*, *L. guyanensis* and *L. panamensis* with both intradomiciliar and peridomiciliar transmission [1,5]. The largest outbreak of CL occurred during 2005 to 2009, with more than 35 000 cases reported in a military population, 80% of which were caused by *L. braziliensis* and the remaining 20% by *L. panamensis*. Another important outbreak, which affected the civilian population, occurred in the Andean valleys in 2003 and 2004, with 2810 cases of CL due to *L. guyanensis* [1].

The vectors responsible for the transmission of leishmaniasis belong to the Phlebotominae subfamily, which includes a wide range of species of medical importance [6]. The subfamily is composed of about 1000 species, and up to 530 species of Phlebotominae from 23 genera have been reported to occur in 28 countries of the New World [7]. The subtribe Lutzomyiina is distributed throughout most of the Americas, although it prefers areas less than 3200 m.a.s.l. and, in some of these countries, its presence is based on climatic variations and/or seasons [7,8]. In Colombia, 163 species of the Phlebotominae subfamily have been described, of which 21 are of medical importance [6,7,9].

In Colombia, one important transmission focus encompasses mainly rural areas of the municipality of Arboledas in the state of Norte de Santander [10]. Between 2014 and 2016, Norte de Santander recorded unstable incidence rates (15.4 ± 51.6 cases per 100 000 inhabitants) with a tendency towards the higher values [9]. Arboledas contributed to 21% (398/1852) of the total cases reported in Norte de Santander during this period. The major *Leishmania* species known to be circulating in Arboledas is *L. braziliensis* [11], but three others have been recorded in the state: *L. braziliensis*, *L. panamensis* and *L. mexicana* [12]. A total of 42 phlebotomine species have been reported in Norte de Santander [13,14], including species of Phlebotominae that are of medical importance [15–17]. This state demands special attention because it has had one of the highest rates of human immigration in recent years, probably as a consequence of the sociopolitical situation in Venezuela [18].

The circulation of a variety of *Leishmania* species and vectors in the border regions of Colombia has been previously studied [6,9]. Epidemiological studies have been conducted in the region since the 1990s. However, the region has undergone remarkable changes in terms of the socioeconomic conditions, human migration and environmental factors, all of which are known to affect species distribution, parasites, reservoirs and hosts in areas of CL transmission. Therefore, it is necessary to continue conducting detailed ecological studies to understand the complexity of these changing scenarios and because, despite the prevention and control activities carried out by public health institutions,

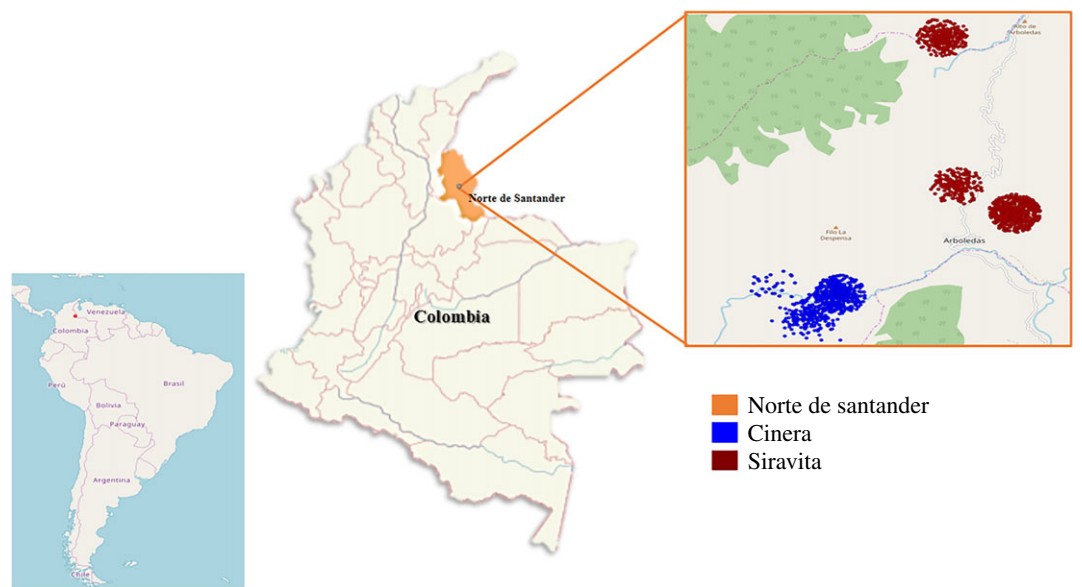

**Figure 1.** Geographical distribution of phlebotomine species collected in this study. The map was constructed using eSPATIAL online.

transmission cycles have been maintained and have intensified over time [9]. The aim of this study was to determine the ecological interactions among sand flies, parasites, and hosts in two localities in Arboledas using a new approach. This consisted of obtaining information on *Leishmania* species and hosts from all collected sand flies with blood in their abdomens. We additionally assessed whether the diversity of Phlebotominae species change across environmental gradients. In this way, we aimed to broaden our understanding of the interactions between parasites, phlebotomines and hosts in Arboledas and to demonstrate a method of study complementary to the ecological approach in areas of CL transmission.

# 2. Methods

## 2.1. Study area

Fieldwork was carried out at two different localities, Cinera and Siravita, in the municipality of Arboledas of the state of Norte de Santander (figure 1). According to the life zones defined by Holdridge [19], this region is characterized by premontane wet forest with mean annual temperatures of 18–24°C, relative humidity of up to 90%, rainfall of up to 2000 mm yr$^{-1}$ and elevations of between 1000 and 2000 m.a.s.l. The rainfall pattern is bimodal, with two rainy seasons with the highest rainfall in March to May and September to November. The lowest rainfall occurs during the months of January, June, July and August [20]. The main economic activity in the region involves coffee plantations, but other products, such as sugar cane, bananas and citrus fruits, are starting to replace the traditional crop. In addition, the region also participates in the production of breeding lines of farm livestock. The area was selected for sampling based on previous studies and casuistic approaches, which have revealed a considerable degree of leishmaniasis prevalence [10]. Universidad del Rosario provided the field permit from ANLA (Autoridad Nacional de Licencias Ambientales) 63257-2014. All collections were done on public land.

## 2.2. Collection of phlebotomines

Entomological sampling was performed over three consecutive nights during the months of June, July and September 2018. To collect sand flies, 18 white-light Centers for Disease Control and Prevention (CDC) traps were installed 50 m above the ground at 18.00 and removed at 06.00 the following morning, resulting in approximately 12 h of exposure per trap. A total of nine CDC light traps were used per night at each location. Traps were distributed throughout diverse environments (remnant forest, coffee plantations, sugar cane plantations, citrus plantations and grassland), identified through the heterogeneity of the landscape. Shannon traps were also set in peridomiciles each night, from 18.00 to 06.00.

During trap collection, sand flies were separated from other dipterans based on their external morphological characteristics and preserved in 1.5 ml plastic vials containing 70% ethanol. Engorged females (with blood visible inside their digestive tracts) were identified and stored in 1.5 ml plastic vials containing isopropanol. On arriving at the laboratory facilities, all samples were stored at 4°C until they were processed.

Taxonomic identification was carried out using Young & Duncan [21] and Galati [22–24] as references. Sand flies were treated with 15% sodium hydroxide for 12 h then 50% acetic acid for 15 min. The samples were subsequently exposed to successive concentrations of alcohol (70%, 80%, 90% and absolute alcohol) for 10 min each. Finally, females and males were embedded in vials containing eugenol to help visualize the internal structures. The phlebotomine sand fly species reported herein follow the nomenclature system of Galati [22] and the generic abbreviations proposed by Marcondes [25].

## 2.3. Molecular analysis of engorged female phlebotomines

We collected 144 female phlebotomines with blood visible inside their digestive tracts and conducted DNA extraction of the whole bodies using the ZR Tissue & Insect Miniprep DNA Zymo™ kit (Zymo Research, Irvine, USA). DNA detection was performed to identify Phlebotominae feeding sources and *Leishmania* species using PCR, followed by the sequencing of various molecular markers, as described below.

### 2.3.1. *Leishmania* DNA detection

DNA from *Leishmania* was detected by amplifying the gene encoding HSP70 using the primers HSP70F (5′-AGGTGAAGGCGACGA-3′) and HSP70R (5′-CGCTTGTCCTTTGCGTC-3′) under the amplification conditions described by Patino *et al.* [26].

### 2.3.2. PCR for feeding-source identification

A 215 bp fragment of the 12S gene was amplified using Go Taq Green Master Mix (Promega), water and the primers L1085 (10 nM) (5′-CCCAAACTGGGATTAGATACCC-3′) and H1259 (10 nM) (5′-GTTTGCTGAAGATGGCGGTA-3′), as described by Velásquez-Ortiz *et al.* [27].

### 2.3.3. PCR for phlebotomine barcoding

The DNA barcode region from the cytochrome oxidase I gene (*COI*) was amplified by PCR. For the PCR reaction, 1 µl total DNA was mixed with the following reagents: 12.5 µl 2 × GoTaq Green Master Mix and 10 µM forward and reverse primers LCO1490 (5′-GGTCAACAAATCATAAAGATATTGG-3′) and HCO2198 (5′-TAAACTTCAGGGTGACCAAAAATCA-3′) [28], to a final reaction volume of 25 µl. The thermocycling conditions consisted of one cycle of 1 min at 94°C, 40 cycles of 40 s at 94°C, 40 s at 52°C, 1 min at 72°C, and finally 5 min at 72°C. The amplified products of *COI*, *HSP70* and *12S* were visualized using 2% agarose gel stained with SYBR safe (Invitrogen).

### 2.3.4. Sequence analysis

PCR products were purified using EXOSAP (Affymetrix, USA) and sequenced by the dideoxy-terminal method in an automated sequencer (AB3730, Applied Biosystems). The sequences were read, edited and aligned using MEGA X. The sequences were also subjected to similarity analysis with sequences deposited in the database using BLASTn. Multiple alignments were built using MUSCLE with default parameters and performed for *HSP70* sequences from *L. braziliensis* and *L. panamensis* and *COI* sequences from *Pi. ovallesi*, *Lu. gomezi* and *Ps. davisi*. Then, for each *Leishmania* and phlebotomine species, a Nexus matrix was constructed for haplotype network analysis in Network 5.0 (https://www.fluxus-engineering.com/sharenet.htm) using a median-joining model based on 1000 iterations with default parameters. For *Leishmania* species, we analysed the distribution of haplotypes at the geographical level (locality for *L. braziliensis*/*L. panamensis* and house number for *L. braziliensis*).

## 2.4. Data analysis

Hill numbers include the three most widely used species diversity measurements. *Species richness* ($q = 0$) is completely insensitive to species abundance and is known as the species richness of an assemblage. *Shannon diversity* ($q = 1$) counts individuals equally, and this measurement thus counts species in

proportion to their abundance. The measurement must be interpreted as the effective number of 'common' species in the assemblage. *Simpson diversity* ($q = 2$) measures where most weight is placed on the most dominant species [29,30]. In this study, we initially obtained the sample completeness for each order of diversity ($q = 0$, 1 and 2) as a measure of the representativeness of the inventory of phlebotomine sand fly species in the study area, locations and in comparisons between environments. Subsequently, we performed an asymptotic analysis to estimate the diversity profiles (Hill numbers of order $q = 0$, 1 and 2) for each location and for comparisons between different environments (forest remnants, coffee plantations, cane plantations, citrus plantations and grassland). For the analysis of diversity and coverage of phlebotomines in the study areas and localities, we included all specimens collected during the fieldwork ($n = 1729$). In the case of the comparative analysis between environments, only those insects collected in CDC light traps ($n = 1317$) were used. The information used to perform the analysis consisted of individual-based abundance data. We extrapolated these data to double the reference sample size, and the base sample size was double the smallest reference sample size to compare diversities [30]. Bootstrap replications for computing the confidence intervals in each diversity parameter were equal to 100, and the level for the confidence intervals was 0.95 [31]. We used the R package iNterpolation/EXTrapolation (iNEXT) to provide functions to compute and plot seamless rarefaction and extrapolation sampling curves for Hill numbers (species richness, Shannon diversity and Simpson diversity) [31]. The proportion of each phlebotomine species was calculated according to sex, location and environment, and the altitude medians were calculated for the phlebotomine collection sites. The $Z$-test and the $\chi^2$-test were used to compare proportions. Non-parametric statistical tests were used to compare numerical variables, and the $p$-values were estimated using Monte Carlo simulations if any variable presented incomplete data (significance level $p < 0.05$).

### 2.4.1. Analysis of interactions among feeding sources, *Leishmania* and phlebotomine species

A mesh chart was created using SPSS version 18 and used to explore the relationships among *Leishmania* species, phlebotomines and feeding sources. Circos plots were used to show specific associations and their frequencies between *Leishmania*/phlebotomine species, *Leishmania* species/feeding sources and phlebotomine species/feeding sources. Finally, a Bayesian model using a Markov Mantle as the structure and a $\chi^2$-test to calculate significance ($p < 0.05$) were used to explore these specific associations.

## 3. Results

### 3.1. Ecology of sand flies

During the sampling period (June, July and September 2018), we collected a total of 1729 specimens, comprising 18 species, of Phlebotominae sand flies. Of these sand flies, 62.3% (821 females and 255 males) were collected from the Siravita locality and 37.7% (558 females and 94 males) from Cinera (figure 1; electronic supplementary material, table S1). The most abundant species in the study area were *Pi. ovallesi* (29.8%), *Ps. davisi* (20.3%), *Pi. spinicrassa* (18.5%) and *Lu. gomezi* (15.8%). The highest species diversity was estimated in Siravita (20 CI 95% 15.94–30.95 versus 15 CI 95% 11.17–21.68), corresponding to the order $q = 0$, but there were no statistically significant differences between the localities. However, for orders $q = 1$ and $q = 2$, Cinera showed a higher effective number of species, which was statistically significant. The values of estimated sample coverage were equally high for all samples (electronic supplementary material, table S1).

The expected coverage-based rarefaction and extrapolation as a function of sample number for Phlebotominae sand flies based on Hill numbers ($q = 0$, 1 and 2) for five environments (forest remnants, coffee plantations, cane plantations, citrus plantations and grassland) were 98, 99, 98, 98 and 99, respectively (electronic supplementary material, table S2). These results indicate that the amount of sampling was sufficient for all sites and that sample completeness allowed valid comparisons to be made based on the estimated Hill numbers.

A greater number of phlebotomines were collected from forest remnants and coffee plantations (electronic supplementary material, table S2). Differences between the diversity estimated for the order $q = 0$ were not statistically significant across the different environments evaluated (electronic supplementary material, table S2). However, the diversity of order $q = 1$ made it possible to detect statistically significant differences between the diversity estimated for grassland and the citrus plantations, coffee plantations and forest remnant environments but not for sugar cane plantations. In the grassland, only 66.8%, 67.2%, 78.1% and

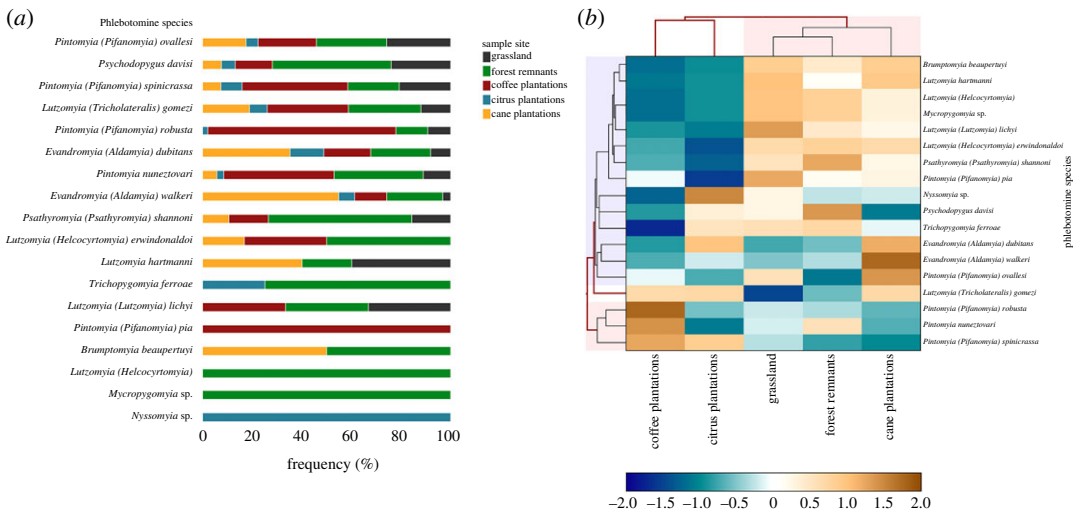

**Figure 2.** Distribution of phlebotomine species by collection environment. (*a*) Distribution of phlebotomines in each environment. (*b*) Clusters of phlebotomine species influenced by the environment.

78.4% of the diversity present in the citrus plantations, coffee plantations, forest remnants and cane plantations, respectively, were observed (electronic supplementary material, table S2).

When using the estimated diversity of order $q = 2$, all sites showed a lower number of effective species, as this measurement focuses on dominant species only. Nonetheless, the significant differences between grassland, coffee plantations and citrus plantations cultivation were maintained, but the differences between cane plantations and forest remnants were not (electronic supplementary material, table S2). Therefore, considering the diversity of the orders $q = 1$ and 2, grassland was the least diverse site because the most abundant species (*Pi. ovallesi*, *Ps. davisi*, *Pi. spinicrassa* and *Lu. gomezi*) represented more than 90% of the specimens collected, reflecting a lower equity in the distribution of abundance of the common species.

In terms of sand fly diversity (figure 2*a*; electronic supplementary material, table S2), the frequencies of the different species of phlebotomines varied based on the locality and environment ($p < 0.001$; X2, Monte Carlo 5000 simulations) (figure 2*a*; electronic supplementary material, table S1). Some species were distributed across all environments, e.g. *Ev. walkeri*, *Pi. nuneztovari*, *Ev. dubitans*, *Pi. spinicrassa*, *Lu. gomezi*, *Ps. davisi* and *Pi. ovallesi* (figure 2*a*). Some species were more abundant in specific environments, e.g. *Evandromyia (Aldamyia) walkeri* in sugar cane plantations, *Pi. spinicrassa* and *Pintomyia (Pifanomyia) robusta* in coffee plantations, and *Ps. davisi* and *Ps. shannoni* in forest remnants (figure 2*b*). There were also species whose collection was limited to specific environments, such as *Nyssomyia* sp. in citrus plantations and *Pintomyia (Pifanomyia) pia* in coffee plantations (figure 2*a*). Figure 2*b* shows that three environments (forest remnants, grassland and sugar cane plantations) were the most similar in terms of phlebotomine sand fly species, and these species form a block in the upper-right corner of the figure. A second block of species, comprising *Ev. dubitans*, *Ev. walkeri*, *Pi. ovallesi* and *Lu. gomezi*, was observed in sugar cane plantations, and a third block in coffee plantations consisting of *Pintomyia* spp. A higher frequency of females than males was observed, and this was statistically significant for *Pi. ovallesi*, *Pi. spinicrassa*, *Lu. gomezi*, *Ps. davisi*, *Pi. robusta* and *Pi. nuneztovari* ($p < 0.05$; Z-test was used to compare proportions).

The median altitudes of sample collection sites were also compared. In Cinera, a greater abundance of the species *Pi. robusta*, *Pi. nuneztovari* and *Pi. spinicrassa* were identified at a higher altitude compared with other species ($p < 0.05$; Kruskal–Wallis test). This was related to the collection site, given that these three species were collected more than 70% of the time at altitudes higher than 1350 m.a.s.l. on coffee plantations (60.7%), intradomicile (9.5%), sugar cane plantations (3.6%) and grassland (2.4%) (data not shown) (electronic supplementary material, figures S1 and S2).

## 3.2. Molecular analysis of engorged females

Females were visually inspected for having blood-fed, and 144 engorged females were analysed using molecular methods for phlebotomine barcoding, *Leishmania* detection, species identification and

**Table 1.** Frequency of feeding sources and *Leishmania* species in engorged phlebotomine females.

| feature | n = 144 (%) | 95% CI | *Leishmania* DNA n (%)[a] |
|---|---|---|---|
| location | | | |
| Siravita | 125 (86.8) | 80.2–91.8 | 67/822 (8.1) |
| Cinera | 19 (13.2) | 8.1–19.8 | 13/558 (2.3) |
| environment | | | |
| sugar cane plantation | 60 (41.7) | 33–55.2 | 25/155 (16.2) |
| coffee plantation | 24 (16.7) | 11.0–23.8 | 14/283 (4.9) |
| remnant forest | 2–3 (16) | 10.4–23.0 | 16/309 (5.1) |
| grassland | 21 (14.6) | 9.3–21.4 | 13/192 (6.7) |
| citrus plantation | 10 (6.9) | 3.4–12.4 | 7/62 (11.2) |
| intradomicile | 6 (4.2) | 1.5–8.8 | 5/20 (25.0) |
| Phlebotominae species | | | |
| *Pintomyia (Pifanomyia) ovallesi* | 52 (36.1) | 22.3–44.5 | 22/448 (4.9) |
| *Lutzomyia (Tricholateralis) gomezi* | 24 (16.7) | 11.0–23.8 | 17/234 (7.3) |
| *Psychodopygus davisi* | 16 (11.1) | 6.5–17.4 | 7/228 (3.0) |
| *Evandromyia (Aldamyia) walkeri* | 15 (14.4) | 6.0–16.6 | 6/18 (33.3) |
| *Pintomyia (Pifanomyia) spinicrassa* | 14 (9.7) | 5.4–15.8 | 11/274 (4.0) |
| *Pintomyia (Pifanomyia) robusta* | 9 (6.3) | 2.9–11.5 | 7/42 (16.6) |
| *Lutzomyia hartmanni* | 5 (3.5) | 1.1–7.9 | 3/5 (60.0) |
| *Lutzomyia (Helcocyrtomyia)* sp. | 3 (2.1) | 0.4–6.0 | 3/3 (100) |
| *Pintomyia nuneztovari* | 3 (2.1) | 0.4–6.0 | 2/41 (4.8) |
| *Nyssomyia* sp. | 2 (1.4) | 0.2–4.9 | 1/2 (50.0) |
| *Psathyromyia (Psathyromyia) shannoni* | 1 (0.7) | 0.02–3.8 | 1/36 (2.7) |
| feeding source | | | |
| *Homo sapiens* | 29 (50) | 36.8–63.4 | |
| *Bos taurus* | 8 (13.8) | 6.1–25.4 | |
| *Canis lupus familiaris* | 6 (10.3) | 3.9–21.1 | |
| *Equus caballus* | 6 (10.3) | 3.9–21.1 | |
| *Sus scrofa* | 5 (8.6) | 2.9–19.0 | |
| *Icterus mesomelas* | 3 (5.2) | 1.1–14.4 | |
| *Gallus gallus* | 1 (1.7) | 0.04–9.2 | |
| *Leishmania* species | | | |
| *Leishmania braziliensis* | 41 (63.1) | 50.2–74.7 | |
| *Leishmania panamensis* | 12 (18.5) | 9.9–30.0 | |
| *Leishmania infantum* | 8 (13.2) | 5.5–22.8 | |
| *Leishmania amazonensis* | 4 (6.1) | 1.7–15.0 | |

[a]Frequency of *Leishmania* DNA detection in all female phlebotomines.

feeding-source characterization. Most specimens were from either sugar cane plantations or coffee plantations in Siravita (table 1).

### 3.2.1. Phlebotomine barcoding

Eleven species of phlebotomines were detected in the samples analysed (table 1). The most frequent phlebotomine species were *Pi. ovallesi* with 52 (36.1%) specimens, *Lu. gomezi* with 24 (16.7%) and *Ps. davisi* with 16 (14.4%) (table 1). The *COI* sequences were deposited in GenBank with the

R. Soc. Open Sci. **7**: 200266

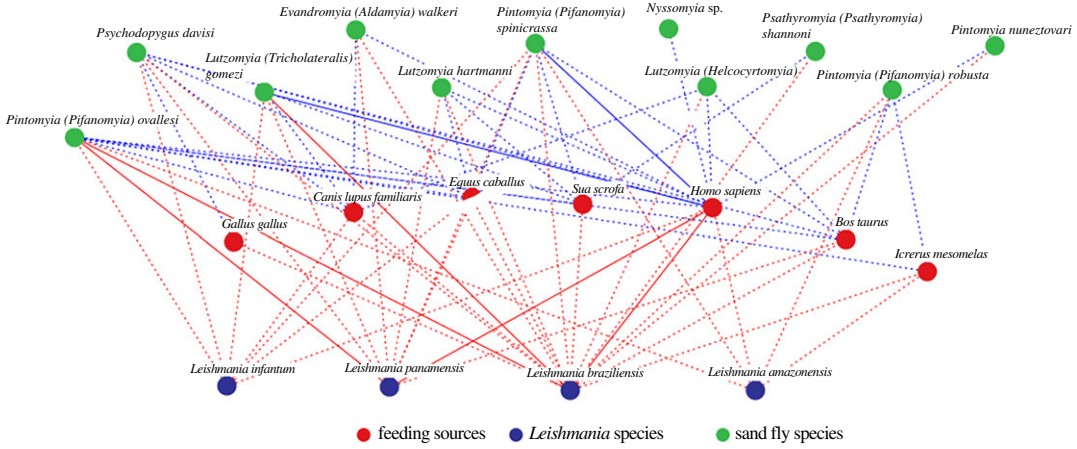

**Figure 3.** Interactions among feeding sources, *Leishmania* and sand fly species.

submission codes grp 7564999, grp 7565144, grp 7565112, grp 7565075, grp 7565068, grp 7565112, grp 7565005 and grp 7565038.

### 3.2.2. *Leishmania* DNA detection and species identification

*Leishmania* DNA was detected in 5.8% (80/1380) of females analysed and corresponded mainly to *Pi. ovallesi* (22/80; 27.2%), *Lu. gomezi* (17/80; 21.3%) and *Pi. spinicrassa* (11/80; 13.8%). *Leishmania* DNA detection frequencies were higher in Siravita and intradomicile environments; however, the differences were not significant. *Leishmania* DNA detection frequencies by location, species and environment are shown in table 1.

We identified four *Leishmania* species from 65 samples. The species *L. braziliensis* and *L. panamensis* showed the highest frequencies, at 63.1% and 18.5%, respectively. The *HSP70* gene sequences were deposited in GenBank with the accession numbers MT096048–MT096102 and MT108453–MT108464.

### 3.2.3. Feeding-source characterization

This analysis was performed for phlebotomines in which *Leishmania* DNA had been detected. The most frequent feeding preferences for females were *Homo sapiens* (50%) and domestic animals (*Bos taurus*, *Canis lupus familiaris* and *Equus caballus*) (table 1). The sequences were deposited in GenBank with the accession numbers MT048587–MT048590, MT048550–MT048586, MT04853, MT048500–MT048507, MT048520–MT048523 and MT048525–MT048530.

### 3.2.4. *Leishmania*, phlebotomine and feeding-source interactions

The relationship analysis showed numerous interactions between *Leishmania* species, phlebotomines and food preference species (figure 3). *Pintomyia spinicrassa*, *Ps. davisi* and *Pi. ovallesi* showed a greater number of interactions with *Leishmania* species and feeding sources. *Leishmania braziliensis* showed a higher number of interactions with phlebotomines and feeding sources. The feeding sources with the greatest number of interactions were *Homo sapiens* and *Bos taurus*. However, stronger relationships were seen among the phlebotomine species *Lu. gomezi* and *Pi. spinicrassa*, the preferred *H. sapiens* food source and *Leishmania* species (*L. braziliensis*/*L. panamensis*) (figure 3).

The interactions between food preferences and phlebotomine species showed that human blood was found in most species (7/9) and most frequently in *Pi. ovallesi*, *Ev. walkeri*, *Pi. spinicrassa* and *Lu. gomezi* (figure 4a). Its associations with the last two were the strongest, as evidenced by Bayesian analysis (figure 5c). *Bos taurus* and *Canis lupus* were the food preferences of several phlebotomine species (figure 4a). *Pintomyia ovallesi* fed on the greatest number of food sources (6/7), with the highest frequency being human and *Bos taurus* blood (figure 4a), and the association with the latter and *Equus caballus* was greater (figure 5c).

Interactions between species of *Leishmania* and phlebotomines were also explored. *Leishmania braziliensis* was found in the majority of the phlebotomine species (10/11) and most frequently in *Pi. ovallesi* and *Lu. gomezi* (figure 4b). *Leishmania panamensis* also showed several interactions with

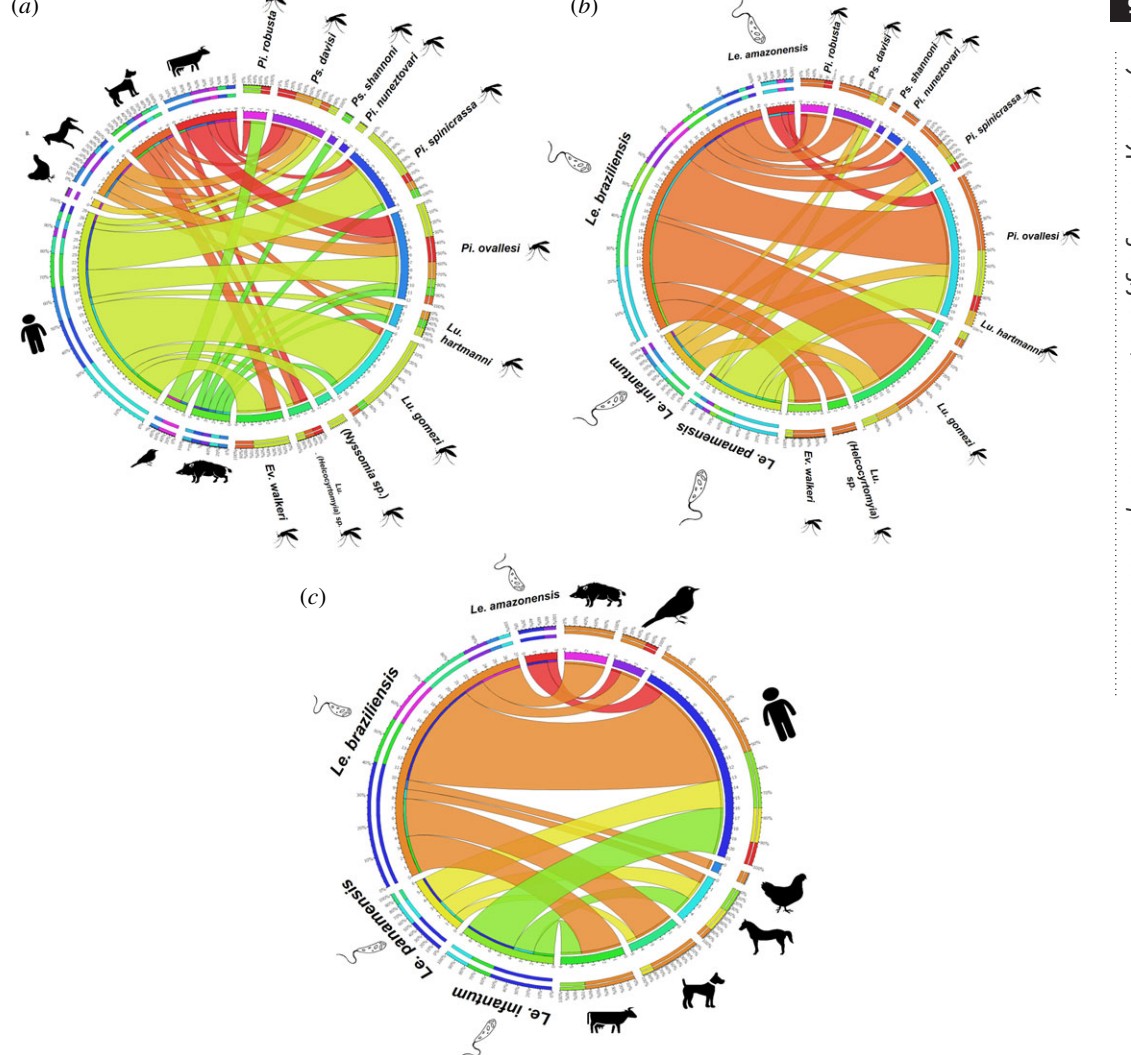

**Figure 4.** Circos plot of feeding source, *Leishmania* and sand fly species relationships. (*a*) Relationships and frequencies between phlebotomine species and feeding sources. (*b*) Relationships and frequencies between *Leishmania* and phlebotomine species. (*c*) Relationships and frequencies between *Leishmania* species and feeding sources.

different species of phlebotomines and, as with *L. braziliensis*, the frequencies were higher in *Pi. ovallesi* and *Lu. gomezi* (figure 4*b*). It was also striking that *L. infantum* interacted with four phlebotomine species (figure 4*b*). Of the three *Leishmania* species mentioned, the strongest associations were seen with *Pi. ovallesi* and *Lu. gomezi*, while *L. amazonensis* showed stronger associations with *Pi. robusta* and *Pi. spinicrassa* (figure 5*b*).

When we analysed the interactions between food preference and *Leishmania* species, *L. braziliensis* again showed the greatest number of interactions with all food sources, with the most frequent being *H. sapiens*, *C. lupus* and *E. caballus*. Whereas, *L. panamensis* and *L. infantum* had the same number of interactions with different food sources and exhibited higher associations with *H. sapiens*, *B. taurus* and *E. caballus* (figures 4*c* and 5*a*). Additionally, there were greater associations between human blood and the four *Leishmania* species found, that had greater associations with domestic animals (*B. taurus*, *C. lupus* and *E. caballus*) (figure 5*a*). Finally, *L. amazonensis* showed interactions with *Icterus mesomelas*, *Sus scrofa* and *H. sapiens* and had the strongest association with *H. sapiens* (figure 5*a*).

### 3.2.5. Genetic diversity of *Leishmania* and phlebotomine species

The genetic diversity of *L. panamensis* and *L. braziliensis* was explored through the construction of haplotype networks, which were analysed by location. In the case of *L. panamensis*, five haplotypes were detected, of which three were found in Siravita (figure 6*a*). In the case of *L. braziliensis*,

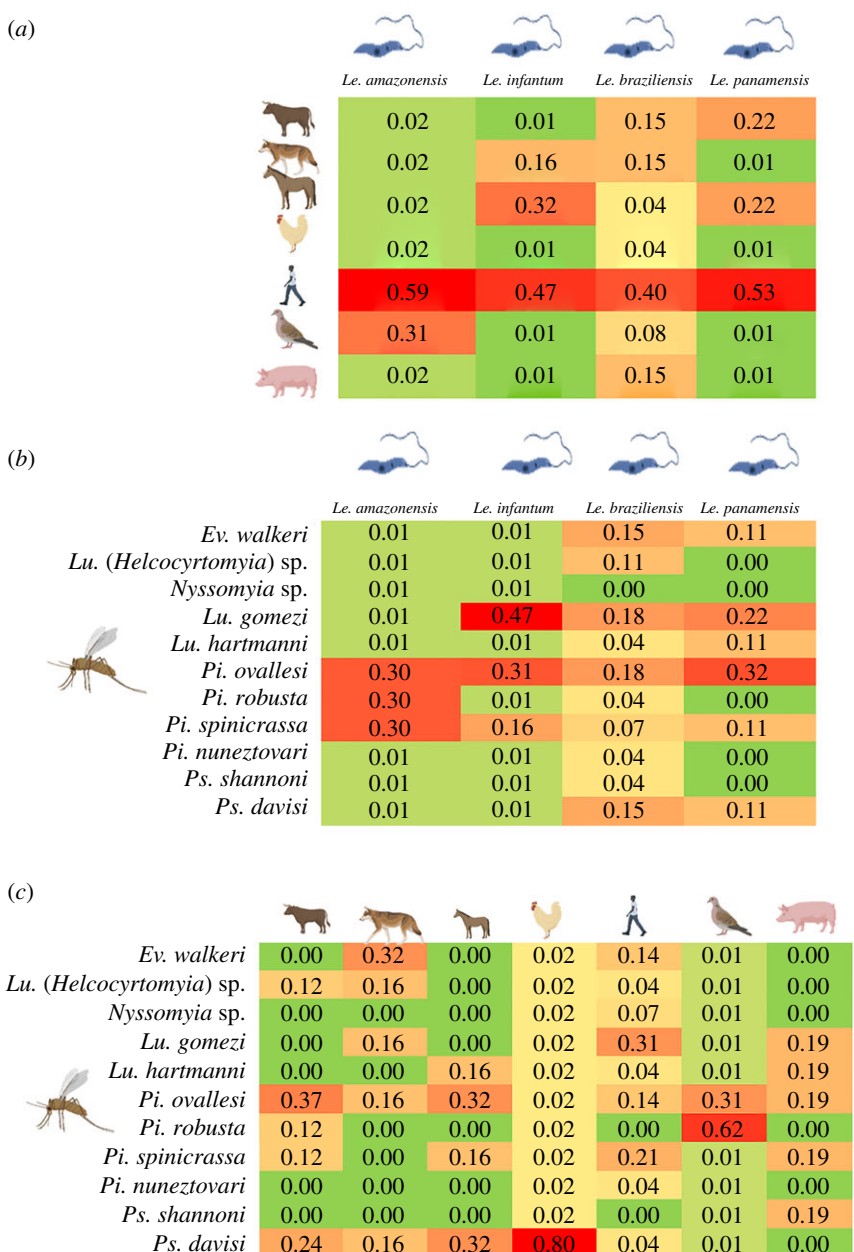

**Figure 5.** Feeding source, *Leishmania* and sand fly species associations based on a Bayesian model. (*a*) Associations between *Leishmania* species and feeding sources. (*b*) Associations and frequencies between *Leishmania* and phlebotomine species. (*c*) Associations between phlebotomine species and feeding sources. Red indicates stronger associations and green indicates weaker associations between variables.

18 haplotypes were found, of which 13 were exclusive to Siravita and three to Cinera (figure 6*b*). These haplotypes were also analysed with respect to the houses where the traps were set up; two haplotypes were found shared between several houses with the remaining haplotypes exclusive to individual houses (figure 6*c*). Finally, haplotype networks for the more frequently occurring phlebotomine species were constructed, and a greater number of *P. ovallesi* haplotypes were detected (electronic supplementary material, figure S3).

## 4. Discussion

Forest remnants, traditional coffee plantations and citrus plantations were found to be the most diverse assemblages in terms of phlebotomine fauna in this study, while grassland and sugar cane plantations were associated with lower-diversity assemblages (diversity of order $q = 1$ and $q = 2$) and were

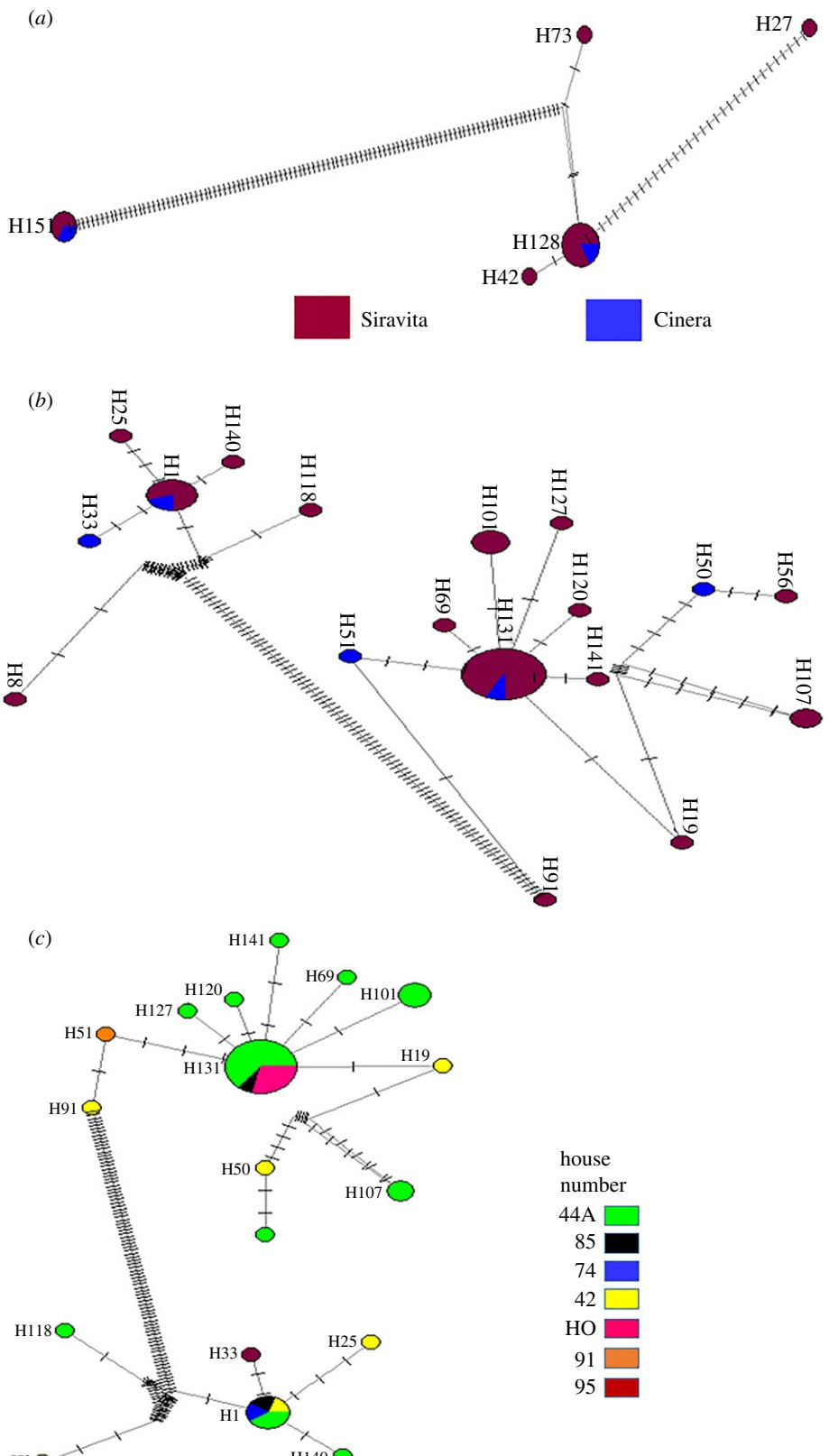

**Figure 6.** Network analysis of the geographical distribution of *Leishmania* species. Alleles of the *HSP70* gene were retrieved to construct the networks shown for each species; the lines specify where a nucleotide change occurred. (*a*) *Leishmania panamensis* haplotypes by location. (*b*) *Leishmania braziliensis* haplotypes by location. (*c*) *Leishmania braziliensis* haplotypes by house.

dominated by species such as *Pi. ovallesi*, *Ps. davisi*, *Pi. spinicrassa* and *Lu. gomezi* (electronic supplementary material, table S2). Some of these species are proven vectors and others are putative vectors of various species of the *Leishmania* genus [15–17,32,33]. In Colombia, Phlebotominae of

medical importance are predominantly distributed within disturbed areas of rural regions where the original vegetation cover has been reduced [6].

Through our sampling efforts, we detected 18 Phlebotomine species, corresponding to 90% of the species predicted to occur in the study area. Previous studies conducted over approximately 4 years in Siravita recorded 17 species of Phlebotominae [10,11]. However, we detected other taxa in the study area (*Lu. scorzai*, *Lu. hartmanni*, *Nyssomyia* sp., *Micropygomyia* sp. and *Pi. robusta*). Apart from *Pi. robusta*, all the species described in this study have been previously reported in Colombia [7,13]. This report was confirmed by *COI* sequencing, with all sequences exhibiting an identity greater than 98% with those previously published for *Pi. robusta* in Peru [34].

It is important to highlight that the species composition recorded in the study area is similar to that reported for a CL focus in the state of Táchira in Venezuela [35–37]. This may be because both endemic areas belong to the orographic complex of the northern Andes and present similar physiographic and ecological characteristics [38] and that *Pi. spinicrassa*, *Pi. nuneztovari*, *Pi. serrana* and *Pi. ovallesi* are all well adapted to the coffee agrosystems of Norte de Santander. Similar trends for these species in terms of elevation have also been recorded in other areas of Colombia, as well as in Bolivia and Venezuela [36,37,39,40]. The changes observed with respect to previous studies, such as the gain and difference in phlebotomine diversity and the occurrence of new species, may be due to changes generated by processes, such as agriculture, domestication, urbanization and climate change, that favour the appearance and/or adaptation of new species and the loss of previously reported species [41]. Nevertheless, sampling should be continued in the region to quantify these shifts in terms of the transmission dynamics of CL.

We consider the preselection of recently fed females an advantage because it provided high sample numbers for the detection of *Leishmania* DNA and, simultaneously, a better understanding of the circulation of the parasites in hosts and vectors. It has been stated that ecological studies of CL and visceral leishmaniasis (VL) foci should include an analysis of insect feeding patterns, as this broadens our epidemiological knowledge of the various interactions between parasites, vectors and reservoirs [42–44].

We discovered that 5.8% of females carried *Leishmania* DNA. This is similar to the results of some studies in Old World settings that found frequencies of between 0.3% to 7.0% in females that had both fed and not fed [44,45]. Our result is in concordance with the findings of several previous studies of vectors in different areas of Brazil where CL is endemic, which described *Leishmania* frequencies of between 3.4% and 8.1% [42,46–48]. In concordance with Souza-Ferreira *et al.* [49], our data also showed there was a large quantity of *Leishmania* DNA in all engorged sand flies sampled. Although *Leishmania* DNA was detected in different phlebotomine species, this does not necessarily reflect infection by the parasite since the females had fed; this was a limitation of our study, as we could not determine the vectorial capacity of the phlebotomines species found in Arboledas. Thus, of the nine phlebotomine species in which *Leishmania* DNA was detected (table 1), two, *Ev. walkeri* and *Pi. robusta*, have never been previously implicated in *Leishmania* transmission [50,51]. Further studies are needed to determine how the presence of *Leishmania* DNA correlates with transmission of the parasite.

In the present study, we described four *Leishmania* species, the most frequent being *L. braziliensis* followed by *L. panamensis*. These results agree with the findings of Young *et al.* [11] and Alexander *et al.* [10], who characterized the study area as a transmission focus of *L. braziliensis*. Previous reports of CL outbreaks in Colombia have described higher frequencies of *L. panamensis* than *L. braziliensis*, with the latter mainly found in sylvatic environments [12,52–55]. As was found in previous studies, we observed a greater genetic diversity in *L. braziliensis* than in *L. panamensis* [9,12]. In addition, some haplotypes were typical for each locality studied (figure 6*a* and *b*) or even more restricted areas, i.e. houses (figure 6*c*). Our results are in accordance with a study carried out by Patino *et al.* [56] in which a greater frequency of *L. braziliensis* was observed among a military population; this was attributed to the contact the soldiers had with enzootic cycles in Colombia. However, in the CL focus of our study, *L. braziliensis* was found to interact with domestic hosts (figure 5*a*, *H. sapiens*, *B. taurus* and *C. lupus*) and with phlebotomines of medical importance that are usually detected in CL foci (figure 5*b*: *Lu. gomezi* and *Pi. ovallesi*). This may be explained by several factors and by taking into account the high genetic diversity observed in *L. braziliensis* and even the appearance of haplotypes as a result of their geographical distribution; it is possible that these genetic changes allowed these species to adapt to new hosts or vectors in domestic settings [9,12,56]. This concurs with existing reports from the San Cristobal municipality of the state of Táchira, Venezuela, which borders Norte de Santander and has the highest number of CL cases in Venezuela, in which *L. braziliensis* has been detected in patients and vectors [1,35–37,57–59].

In agreement with studies of CL outbreaks previously conducted in Córdoba, Colombia [17,42], food sources were primarily of domestic origin, the most frequent being human blood (table 1). This striking finding illustrates the considerable risk of *Leishmania* transmission to humans and accounts for the high number of cases in the region. Human blood was found in seven phlebotomine species (figures 4*a* and 5*c*), which emphasizes the medical importance of these vectors and their possible role in the transmission of the parasite in Arboledas. Cows, canines and horses were also found to participate in transmission, and their roles in transmission dynamics are likely to be highly relevant given their proximity to humans. The infection of canines in Colombia with various *Leishmania* species has been described previously [9,12,60]; clearly, they have important roles in *Leishmania* transmission dynamics. However, it is not apparent if they act as barriers, preventing infection in humans (dilution effect), or if they act as reservoirs or sources of the parasite [60,61]. The detection of blood from cows, horses and pigs is also very important, especially with regard to the roles they play in the domestic environment, their mobility between localities and environments, and their proximity to humans. In Brazil and Venezuela, horses and pigs were also found to be infected and to show clinical manifestations [62–66]. These findings are an indication of the complexity of *Leishmania* transmission dynamics in the region and the need for future studies.

Network analysis (figure 3) showed the important connections between the phlebotomines, *Pintomyia ovallesi* and *Lu. gomezi,* and the parasites *L. braziliensis* and *L. panamensis.* This result indicates the close ecological interactions among the two *Leishmania* species and the phlebotomines species, which coincide with the high percentages of distribution for these two parasites with *Pi. ovallesi* (63% *L. braziliensis* and 73% *L. panamensis*) and *Lu. gomezi* (52% *L. braziliensis* and 66% *L. panamensis*) [55]. Colombian populations of *Pi. ovallesi* from a coffee area in Cundinamarca state, which is endemic for CL, were susceptible to infection with *L. braziliensis* under laboratory conditions [67]. In Guatemala and Venezuela, *Pi. ovallesi* is a proven vector of *L. braziliensis* and *L. mexicana* [68,69]. The ability of *Pi. ovallesi* to feed from different hosts in the study area, along with the limited variation in its abundance among the dissimilar environments in which it was found, are suggestive of the vector's successful adaptation to transformed environments. These results concur with the high prediction values between the distribution of *Pi. ovallesi* in transformed environments, such as agro-ecosystems (sugar cane and oil palm plantations), estimated for the country [51] and the diversity of this species in the study area (electronic supplementary material, figure S3). In other regions of Colombia, *Lu. gomezi* has been identified as having marked anthropophilic behaviour [15,70–72]. In this study, *Lu. gomezi* showed moderate genetic variability (electronic supplementary material, figure S3B), most likely related to the smaller number of interactions.

Although the analysis of the fed females in this study did not allow the vector capacity of the insects to be evaluated, the results provided information on their interactions with human hosts, domestic animals and *Leishmania* species. Thus, the high genetic variability detected in *Pi. ovallesi* (electronic supplementary material, figure S3A) and *L. braziliensis* (figure 6*b* and *c*), and their interactions with human and domestic reservoirs, may agree with the parasite mosaic theory, as this genetic variability could have arisen from parasite-host/host-vector adaptation processes that have not been previously documented in Arboledas [10,11]. Furthermore, this variability can arise from reciprocal selective pressures generated by anthroponotic processes (agriculture, domestication, urbanization, globalization, climate change and the phenomenon of Venezuelan migration).

In conclusion, our findings suggest that CL transmission scenarios in Colombia are more complex than previously thought. It will be necessary to study these phenomena in more detail with more powerful analytic tools. We emphasize this aspect because it is likely that we are facing a plethora of overlapping transmission cycles within the same ecological scenario, even within small geographical areas, as demonstrated in this study. Therefore, a response that prevents and mitigates the risk of CL transmission and allows appropriate diagnosis and management of patients must start from a thorough understanding of all the elements involved in its transmission. A comprehensive characterization of the molecular eco-epidemiology of leishmaniasis is pivotal before control and prevention strategies are planned for the endemic foci of this disease.

Data accessibility. All data from this study are available in full in the electronic supplementary material, including the complete database. The DNA sequences obtained in this study are available under the accession nos. MT096048–MT096102 and MT108453–MT108464.

Authors' contributions. Conceptualization: C.M.S.-R., C.H., R.G.-M. and J.D.R. Formal analysis: C.M.S.-R., C.H., A.A.T. and R.A.M.-V. Funding acquisition: C.M.S.-R., J.D.R. and A.A.-M. Investigation: C.M.S.-R., C.H., D.M. and A.A.-M. Methodology: C.M.S.-R., C.H., A.A.T., R.G.-M., R.A.M.-V., R.H.-L. and J.D.R. Writing—original draft: C.M.S.-R., C.H. and J.D.R. Writing—review and editing: C.M.S.-R., C.H., R.A.M.-V. and J.D.R.

Competing interests. The authors declare that the research was conducted in the absence of any commercial or financial relationships that could be construed as a potential conflict of interest.

Funding. This study was funded by the Universidad de Santander 'Convocatoria Interna Focalizada de Proyectos de Investigación y Desarrollo Tecnológico 2017–2018'. Project no. PICF117207731763EJ and Initiation Act no. 063-17 and Direccion de Investigacion e Innovacion from Universidad del Rosario. C.H. is funded by the Colombian Science, Technology and Innovation Department (Colciencias) call for PhD training in Colombia, within the framework of the National Program for Promoting Research Training (sponsorship call 727).

Acknowledgements. We thank Darío Sarmiento and Edison Devia from Laboratorio Departamental de Salud Pública, and Johana Yañez, Ciro González and Alvaro Rolón from Grupo de control Vectores of Instituto Departamental de Salud de Norte de Santander (IDS) for their assistance in the fieldwork. We thank Dirección de Investigación e Innovación for providing the English editing service of this manuscript. We thank Suzanne Leech, PhD, from Edanz Group (https://en-author-services.edanzgroup.com/) for editing a draft of this manuscript.

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
