## [Reviewer comments · Royal Society Open Science]

Review History

RSOS-200266.R0 (Original submission)

Review form: Reviewer 1

Is the manuscript scientifically sound in its present form?

No

Are the interpretations and conclusions justified by the results?

No

Is the language acceptable?

Yes

Do you have any ethical concerns with this paper?

No

Have you any concerns about statistical analyses in this paper?

No

Recommendation?

Reject

Comments to the Author(s)

In this study, to elucidate transmission dynamics of cutaneous leishmaniasis in a limited endemic area, the authors analyzed ecology and feeding source of sand flies, and *Leishmania* species in sand flies. Although many data were presented and analyzed, it was unfocused, and more importantly, there are many issues for the data collection process that is the key for this study.

1. Detection of *Leishmania* DNA was performed using engorged female sand flies. However, vector research must be done with sand flies without blood in the gut. Vector species support specific parasite growth, in other words, parasites must be detected in the vector even after digestion of host blood. Therefore, *Leishmania* DNA detected in this study possibly derived from host blood at the best.
2. The positive rate of sand flies is impossibly high. I doubt contamination during sample preparation processes. If the authors believe this data is correct, it must be shown by dissection of sand flies. Dissection of only 10 sand flies will be enough if the results are correct.
3. As the ecology of sand flies, the authors focused on only flora of each area. However, temperature and humidity affect more on sand fly ecology. In addition, fauna is another important factor as the feeding source of sand flies, and it must be addressed in this kind of study.
4. Tables 1 and 2 should be moved to supplementary data.
5. Overall, the resolution of figures must be improved.
6. No sequence data was presented. The data must be shown as a supplementary data or, at least, deposited in GenBank.
7. The purpose of barcoding is unclear.
8. Discussion is too long and unfocused.

Review form: Reviewer 2

Is the manuscript scientifically sound in its present form?

Yes

Are the interpretations and conclusions justified by the results?

Yes

Is the language acceptable?

Yes

Do you have any ethical concerns with this paper?

No

Have you any concerns about statistical analyses in this paper?

No

Recommendation?

Accept with minor revision (please list in comments)

Comments to the Author(s)

The manuscript "Complex ecological interactions across a focus of cutaneous leishmaniasis in eastern Colombia" aimed to analyze the patterns of diversity, food preferences, and *Leishmania* species in the sand fly to elucidate the transmission dynamics in one focus of cutaneous leishmaniasis. The study was done in rural areas in the municipality of Arboledas in the state of Norte de Santander, Colombia

It is an elegant and complete study about cutaneous leishmaniasis epidemiology and should be published.

Major concern

The study is very descriptive and did not, in my opinion, show graphically where the transmission cycles overlap. The actual figures (5 and 6) help to understand, but in fact, the transmission cycles could be presented by octopus and not by parasite, reservoir, and vector. What cycles of what species of parasites are contained in an area. The authors limit themselves to saying that transmission scenarios in the country are more complex than previously thought. These are overlapping transmission cycles within the same ecological scenario. I was hoping that at the end of the manuscript, the authors would give an overview of what is happening. Would not there be a host or vector specificity? Would there be an anthropization of cutaneous leishmaniasis? Would the parasite mosaic theory or the Stockholm paradigm explain what is happening?

Minor concerns

- 1- In the introduction, the authors give a number that needs to be revised. According to Alvar and colleagues, 98 countries, 3 territories and 5 continents are affected by leishmaniasis (Alvar et al 2012). The authors state that according to the World Health Organization (WHO), leishmaniasis is present in at least 88 countries in the world and cite Alvar et al, 2012.
- 2- Line 120, please revise. The collection was done in 3 nights and not in 3 days,
- 3- In lines 324 - 326, please delete the first paragraph. It is about materials and methods.
- 4- Inline 179, please delete the bold characters.
- 5- The references do not respect the rules of the nomenclature. The scientific name should be put in italics and the names of journals should be put as the guidelines.
- 6- The sentence of 364 is taken from the authors or from Urrutia et al. 2018?
- 7- Inline 385, what is the L? please give by the extender.
- 8- Inline 387, tables 1 and 2 are correct.
- 9- To what the authors attribute the fact that the authors did not find *Lu longipalpis*. Wouldn't it be because the vector has already been urbanized?
- 10- In line 509, I think that rote transmission is not the best word.

Review form: Reviewer 3

Is the manuscript scientifically sound in its present form?

Yes

Are the interpretations and conclusions justified by the results?

Yes

Is the language acceptable?

Yes

Do you have any ethical concerns with this paper?

No

Have you any concerns about statistical analyses in this paper?

No

Recommendation?

Accept with minor revision (please list in comments)

Comments to the Author(s)

Congratulations on the manuscript and the scientific work. The statistical analysis is strong and makes it possible to establish the conclusions clearly. Nevertheless it is necessary to fit some simple topics.

In line 28 and 36 the term "food preferences" is very broad. There is only evidence of blood intake. Adjustment is required.

In line 29 explain the ecoregion where the focus of leishmaniasis studied is located.

Specify the department where the two locations are located (Summary).

It is possible to better compare the diversity indices between the Localities (Macro approach) and between microenvironments (Micro approach). It is also necessary to place numeric values of the indexes (Summary).

In the line 44, can improve the redation of "circulating feeding sources of domestic animals". The term circulation is confused.

In the introduction it is necessary to include the abbreviation of Leishmania the first time that it is enunciated, to continue using it in the rest of the document.

In the line 97 to complement "Groves, North of Santander"

Update the reference of Galati 2016 and Galati 2003; by the keys of Galati 2018, on line 33 and Linea 38.

Because only visually blood-fed females were included. It is necessary to clarify the selection criteria and the way in which they interpret the result.

Line 169. Include Network 5.0 Reference

Line 174-200: Ecology of sand flies: The text is very extensive, it is suggested to summarize. "Detection of feeding sources". Modify title more specifically.

Decision letter (RSOS-200266.R0)

Dear Dr Ramírez,

The editors assigned to your paper ("Complex ecological interactions across a focus of cutaneous leishmaniasis in eastern Colombia") have now received comments from reviewers. We would like you to revise your paper in accordance with the referee and Associate Editor suggestions which can be found below (not including confidential reports to the Editor). Please note this decision does not guarantee eventual acceptance.

Please submit a copy of your revised paper before 20-May-2020. Please note that the revision deadline will expire at 00.00am on this date. If we do not hear from you within this time then it will be assumed that the paper has been withdrawn. In exceptional circumstances, extensions may be possible if agreed with the Editorial Office in advance. We do not allow multiple rounds of revision so we urge you to make every effort to fully address all of the comments at this stage. If deemed necessary by the Editors, your manuscript will be sent back to one or more of the original reviewers for assessment. If the original reviewers are not available, we may invite new reviewers.

- Data accessibility

If you wish to submit your supporting data or code to Dryad (<http://datadryad.org/>), or modify your current submission to dryad, please use the following link:
<http://datadryad.org/submit?journalID=RSOS&manu=RSOS-200266>

- Competing interests

- Authors' contributions

AB carried out the molecular lab work, participated in data analysis, carried out sequence alignments, participated in the design of the study and drafted the manuscript; CD carried out the statistical analyses; EF collected field data; GH conceived of the study, designed the study,

coordinated the study and helped draft the manuscript. All authors gave final approval for publication.

- Acknowledgements

- Funding statement

Kind regards,

Anita Kristiansen
Editorial Coordinator

on behalf of Dr Krijn Paaijmans (Associate Editor) and Pete Smith (Subject Editor)
openscience@royalsociety.org

Associate Editor's comments (Dr Krijn Paaijmans):

Comments to the Author:

Dear authors,

I agree with reviewers that your study about cutaneous leishmaniasis epidemiology in Colombia is elegant and provides a substantial contribution to our understanding of the interaction between sand flies and blood meal hosts.

The reviewers did, however, raise some concerns that need to be addressed. In particular, the comment of reviewer 1 about the detection of *Leishmania* DNA (using complete sandfly bodies) is worrying me, as parasites may have been present in the host blood meal, which does not mean the sandfly species is a competent vector. If this point cannot be addressed (due to the lack of additional samples for analysis), this caveat should be clearly highlighted in both the results and discussion. The remaining data on densities, blood meal analysis, etc. are still worth publishing (but focus on additional explanatory factors, beyond the flora of each area).

Finally, the length of the paper can be reduced and please provide an explanation at the end of the manuscript (see comments reviewers 2 and 3).

Comments to Author:

Reviewers' Comments to Author:

Reviewer: 1

Comments to the Author(s)

In this study, to elucidate transmission dynamics of cutaneous leishmaniasis in a limited endemic area, the authors analyzed ecology and feeding source of sand flies, and *Leishmania* species in sand flies. Although many data were presented and analyzed, it was unfocused, and more importantly, there are many issues for the data collection process that is the key for this study.

1. Detection of Leishmania DNA was performed using engorged female sand flies. However, vector research must be done with sand flies without blood in the gut. Vector species support specific parasite growth, in other words, parasites must be detected in the vector even after digestion of host blood. Therefore, Leishmania DNA detected in this study possibly derived from host blood at the best.
2. The positive rate of sand flies is impossibly high. I doubt contamination during sample preparation processes. If the authors believe this data is correct, it must be shown by dissection of sand flies. Dissection of only 10 sand flies will be enough if the results are correct.
3. As the ecology of sand flies, the authors focused on only flora of each area. However, temperature and humidity affect more on sand fly ecology. In addition, fauna is another important factor as the feeding source of sand flies, and it must be addressed in this kind of study.
4. Tables 1 and 2 should be moved to supplementary data.
5. Overall, the resolution of figures must be improved.
6. No sequence data was presented. The data must be shown as a supplementary data or, at least, deposited in GenBank.
7. The purpose of barcoding is unclear.
8. Discussion is too long and unfocused.

Reviewer: 2

Comments to the Author(s)

The manuscript "Complex ecological interactions across a focus of cutaneous leishmaniasis in eastern Colombia" aimed to analyze the patterns of diversity, food preferences, and Leishmania species in the sand fly to elucidate the transmission dynamics in one focus of cutaneous leishmaniasis. The study was done in rural areas in the municipality of Arboledas in the state of Norte de Santander, Colombia

It is an elegant and complete study about cutaneous leishmaniasis epidemiology and should be published.

Major concern

The study is very descriptive and did not, in my opinion, show graphically where the transmission cycles overlap. The actual figures (5 and 6) help to understand, but in fact, the transmission cycles could be presented by octopus and not by parasite, reservoir, and vector. What cycles of what species of parasites are contained in an area. The authors limit themselves to saying that transmission scenarios in the country are more complex than previously thought. These are overlapping transmission cycles within the same ecological scenario. I was hoping that at the end of the manuscript, the authors would give an overview of what is happening. Would not there be a host or vector specificity? Would there be an anthropization of cutaneous leishmaniasis? Would the parasite mosaic theory or the Stockholm paradigm explain what is happening?

Minor concerns

- 1- In the introduction, the authors give a number that needs to be revised. According to Alvar and colleagues, 98 countries, 3 territories and 5 continents are affected by leishmaniasis (Alvar et al 2012). The authors state that according to the World Health Organization (WHO), leishmaniasis is present in at least 88 countries in the world and cite Alvar et al, 2012.
- 2- Line 120, please revise. The collection was done in 3 nights and not in 3 days,
- 3- In lines 324 - 326, please delete the first paragraph. It is about materials and methods.
- 4- Inline 179, please delete the bold characters.
- 5- The references do not respect the rules of the nomenclature. The scientific name should be put in italics and the names of journals should be put as the guidelines.
- 6- The sentence of 364 is taken from the authors or from Urrutia et al. 2018?
- 7- Inline 385, what is the L? please give by the extender.
- 8- Inline 387, tables 1 and 2 are correct.

- 9- To what the authors attribute the fact that the authors did not find *Lu longipalpis*. Wouldn't it be because the vector has already been urbanized?
- 10- In line 509, I think that rote transmission is not the best word.

Reviewer: 3

Comments to the Author(s)

Congratulations on the manuscript and the scientific work. The statistical analysis is strong and makes it possible to establish the conclusions clearly. Nevertheless it is necessary to fit some simple topics.

In line 28 and 36 the term "food preferences" is very broad. There is only evidence of blood intake. Adjustment is required.

In line 29 explain the ecoregion where the focus of leishmaniasis studied is located.

Specify the department where the two locations are located (Summary).

It is possible to better compare the diversity indices between the Localities (Macro approach) and between microenvironments (Micro approach). It is also necessary to place numeric values of the indexes (Summary).

In the line 44, can improve the redation of "circulating feeding sources of domestic animals". The term circulation is confused.

In the introduction it is necessary to include the abbreviation of *Leishmania* the first time that it is enunciated, to continue using it in the rest of the document.

In the line 97 to complement "Groves, North of Santander"

Update the reference of Galati 2016 and Galati 2003; by the keys of Galati 2018, on line 33 and Linea 38.

Because only visually blood-fed females were included. It is necessary to clarify the selection criteria and the way in which they interpret the result.

Line 169. Include Network 5.0 Reference

Line 174-200: Ecology of sand flies: The text is very extensive, it is suggested to summarize. "Detection of feeding sources". Modify title more specifically.

Author's Response to Decision Letter for (RSOS-200266.R0)

See Appendix A.

Decision letter (RSOS-200266.R1)

Dear Dr Ramírez:

On behalf of the Editors, I am pleased to inform you that your Manuscript RSOS-200266.R1 entitled "Complex ecological interactions across a focus of cutaneous leishmaniasis in eastern Colombia" has been accepted for publication in Royal Society Open Science subject to minor revision in accordance with the referee suggestions. Please find the referees' comments at the end of this email.

The reviewers and Subject Editor have recommended publication, but also suggest some minor revisions to your manuscript. Therefore, I invite you to respond to the comments and revise your manuscript.

- Ethics statement

- Data accessibility

If you wish to submit your supporting data or code to Dryad (<http://datadryad.org/>), or modify your current submission to dryad, please use the following link:
<http://datadryad.org/submit?journalID=RSOS&manu=RSOS-200266.R1>

- Competing interests

- Authors' contributions

- Acknowledgements

- Funding statement

Please note that we cannot publish your manuscript without these end statements included. We have included a screenshot example of the end statements for reference. If you feel that a given

heading is not relevant to your paper, please nevertheless include the heading and explicitly state that it is not relevant to your work.

Because the schedule for publication is very tight, it is a condition of publication that you submit the revised version of your manuscript before 28-May-2020. Please note that the revision deadline will expire at 00.00am on this date. If you do not think you will be able to meet this date please let me know immediately.

Kind regards,

Anita Kristiansen
Editorial Coordinator

on behalf of Dr Krijn Paaijmans (Associate Editor) and Pete Smith (Subject Editor)
openscience@royalsociety.org

Reviewer comments to Author:

Please have the manuscript read through by a native English speaker to check / improve the language where necessary.

Author's Response to Decision Letter for (RSOS-200266.R1)

See Appendix B.

Decision letter (RSOS-200266.R2)

Dear Dr Ramírez,

It is a pleasure to accept your manuscript entitled "Complex ecological interactions across a focus of cutaneous leishmaniasis in eastern Colombia" in its current form for publication in Royal Society Open Science.

on behalf of Dr Krijn Paaijmans (Associate Editor) and Pete Smith (Subject Editor)
openscience@royalsociety.org

Appendix A

Bogotá D.C. May 7th, 2020

Doctors

Krijn Paaijmans (Associate Editor)

Pete Smith (Subject Editor)

Dear Doctors,

First of all, we want to thank your consideration with our manuscript. We have responded and applied the suggestions and comments raised by the reviewers that have substantially improved the quality of our manuscript. These can be found below

Reviewers' Comments to Author:

Reviewer: 1

Comments to the Author(s)

In this study, to elucidate transmission dynamics of cutaneous leishmaniasis in a limited endemic area, the authors analyzed ecology and feeding source of sand flies, and *Leishmania* species in sand flies. Although many data were presented and analyzed, it was unfocused, and more importantly, there are many issues for the data collection process that is the key for this study.

1. Detection of *Leishmania* DNA was performed using engorged female sand flies. However, vector research must be done with sand flies without blood in the gut. Vector species support specific parasite growth, in other words, parasites must be detected in the vector even after digestion of host blood. Therefore, *Leishmania* DNA detected in this study possibly derived from host blood at the best.

Response:

We agree with the reviewer, the main limitation of this study was the selection of engorged females for DNA *Leishmania* detection. However, other studies have used the same methodology (Jaouadi et al., 2018). In addition, the aim of this study was to determine ecological interactions among sandflies, *Leishmania* species, and hosts, for that reason we included only engorged females because we did not search to evaluate the vectorial capacity of Phlebotomine species.

To circumvent this comment, the next paragraph was added in the text:

“Although *Leishmania* DNA was detected in different species of phlebotomines, this detection does not necessarily reflect infection by the parasite since the females had fed. This was a limitation to detect the vectorial capacity of the phlebotomines species found in Arboledas.”

2. The positive rate of sand flies is impossibly high. I doubt contamination during sample preparation processes. If the authors believe this data is correct, it must be shown by dissection of sand flies. Dissection of only 10 sand flies will be enough if the results are correct.

Response

The positive rate of sand flies in this study was high probably because we detected DNA from *Leishmania* only in engorged females. In addition, other studies that include fed and not fed females had reported similar rates: Chargui et al.,2028 (3.4%), Gonzalez et al.,2017 (3.7%), Hashiguchi et al.,2019 (0.75-8.33%), Mhaidi et al.,2018 (2.51%- 7.27%) and Zorrilla et al.,2017 (6-8%). In addition, we used water as a negative control in DNA extraction and PCR assays. These controls were processed, and negative results were obtained in all assays. The dissection of sand flies is not possible because the complete body from insects were extracted. However, as we stated above, we detected *Leishmania* DNA, we never attempted to determine the transmission of the parasite of vectorial capacity of the phlebotomines studied.

3. As the ecology of sand flies, the authors focused on only flora of each area. However, temperature and humidity affect more on sand fly ecology. In addition, fauna is another important factor as the feeding source of sand flies, and it must be addressed in this kind of study.

Response

We agree with the reviewer but unfortunately we did not collect data about temperature, humidity, and fauna during the sampling. In addition, the locality had access problems (guerrillas and paramilitary areas – armed conflict areas) and do not exist official databases about these.

4. Tables 1 and 2 should be moved to supplementary data.

Response

The tables were included in supplementary data as suggested

5. Overall, the resolution of figures must be improved.

Response

The resolution of the figures was improved.

6. No sequence data was presented. The data must be shown as a supplementary data or, at least, deposited in GenBank.

Response

The sequences were deposited in GenBank, the accession numbers are now provided in this version.

7. Discussion is too long and unfocused.

Response

We thank the comment of the reviewer. The discussion was shortened as suggested.

Reviewer: 2

Comments to the Author(s)

The manuscript “Complex ecological interactions across a focus of cutaneous leishmaniasis in eastern Colombia” aimed to analyze the patterns of diversity, food preferences, and Leishmania species in the sand fly to elucidate the transmission dynamics in one focus of cutaneous leishmaniasis. The study was done in rural areas in the municipality of Arboledas in the state of Norte de Santander, Colombia. It is an elegant and complete study about cutaneous leishmaniasis epidemiology and should be published.

Response

We deeply thank the comment of the reviewer

Major concern

The study is very descriptive and did not, in my opinion, show graphically where the transmission cycles overlap. The actual figures (5 and 6) help to understand, but in fact, the transmission cycles could be presented by octopus and not by parasite, reservoir, and vector. What cycles of what species of parasites are contained in an area. The authors limit themselves to saying that transmission scenarios in the country are more complex than previously thought. These are overlapping transmission cycles within the same ecological scenario. I was hoping that at the end of the manuscript, the authors would give an overview of what is happening. Would not there be a host or vector specificity? Would there be an anthropization of cutaneous leishmaniasis? Would the parasite mosaic theory or the Stockholm paradigm explain what is happening?

Response:

We do not have data about the transmission cycles and clinical cases of cutaneous leishmaniasis coordinates in Arboledas. However, we consider that the information of reservoirs, phlebotomines and *Leishmania* species circulating in Arboledas is very important for the study of this focus of cutaneous leishmaniasis but is necessary in the near future studies to collect samples from human cases for understanding the transmission dynamics of *Leishmania* species, for improving and focus the control strategies.

We thank the reviewer about the hypothesis of anthropization and mosaic theory that have been included in the discussion as well.

Minor concerns

1- In the introduction, the authors give a number that needs to be revised. According to Alvar and colleagues, 98 countries, 3 territories and 5 continents are affected by leishmaniasis (Alvar et al 2012). The authors state that according to the World Health Organization (WHO), leishmaniasis is present in at least 88 countries in the world and cite Alvar et al, 2012.

Response

The text was modified

2- Line 120, please revise. The collection was done in 3 nights and not in 3 days,

Response

The text was modified

3- In lines 324 - 326, please delete the first paragraph. It is about materials and methods.

Response

The text was modified

4- Inline 179, please delete the bold characters.

Response

The text was modified

5- The references do not respect the rules of the nomenclature. The scientific name should be put in italics and the names of journals should be put as the guidelines.

Response

The text was modified

6- The sentence of 364 is taken from the authors or from Urrutia et al. 2018?

Response

Deleted as suggested

7- Inline 385, what is the L? please give by the extender.

Response

The text was modified

8- Inline 387, tables 1 and 2 are correct.

9- To what the authors attribute the fact that the authors did not find Lu longipalpis. Wouldn't it be because the vector has already been urbanized?

10- In line 509, I think that wrote transmission is not the best word.

Response

The text was modified

Reviewer: 3

Comments to the Author(s)

Congratulations on the manuscript and the scientific work. The statistical analysis is strong and makes it possible to establish the conclusions clearly. Nevertheless it is necessary to fit some simple topics.

In line 28 and 36 the term "food preferences" is very broad. There is only evidence of blood intake. Adjustment is required.

In line 29 explain the ecoregion where the focus of leishmaniasis studied is located. Specify the department where the two locations are located (Summary).

It is possible to better compare the diversity indices between the Localities (Macro approach) and between microenvironments (Micro approach). It is also necessary to place numeric values of the indexes (Summary).

In the line 44, can improve the redation of "circulating feeding sources of domestic animals". The term circulation is confused.

Response

We thank the valuable comments of the reviewer. The text was modified

In the introduction it is necessary to include the abbreviation of Leishmania the first time that it is enunciated, to continue using it in the rest of the document.

Response

The text was modified

In the line 97 to complement “Groves, North of Santander”

Update the reference of Galati 2016 and Galati 2003; by the keys of Galati 2018, on line 33 and Linea 38.

Response

We added reference in methods

Because only visually blood-fed females were included. It is necessary to clarify the selection criteria and the way in which they interpret the result.

Response

We added the selection criteria in the Discussion

Line 169. Include Network 5.0 Reference

Response

We added the link of the software

Line 174-200: Ecology of sand flies: The text is very extensive; it is suggested to summarize.

“Detection of feeding sources”. Modify title more specifically.

Response

The text was modified

Best regards

The authors

Appendix B

Bogotá D.C. June 1st, 2020

Doctors

Krijn Paaijmans (Associate Editor)

Pete Smith (Subject Editor)

Dear Doctors,

We thank your soon response and positive decision. The manuscript was revised by a native English speaker from the EDANZ GROUP, please find below a screenshot of the certificate:

Certificate of Editing

Edited Provisional Title
Complex ecological interactions across a focus of cutaneous leishmaniasis in Eastern Colombia: Novel description of Leishmania species, hosts and phlebotomine fauna

Client name and Institution
Ana Aldana, Universidad del Rosario

Date Completed 2020-05-29	Identification code 86514
------------------------------

Certificate issued by
Koji Yamashita
Managing Director and CEO

Expert Editor: Suzanne Leech
2003 PhD Molecular Parasitology
Queen's University Belfast
Zoology, Microbiology, Genetics

en-author-services.edanzgroup.com

While this certificate confirms the authors have used Edanz's editing services, we cannot guarantee that additional changes have not been made after our edits.

We hope that this new version meets the required criteria of the journal.

Best

The authors